# Efficient Demulsification of Acidic Oil-In-Water Emulsions with Silane-Coupled Modified TiO$_2$ Pillared Montmorillonite



**Gaohong Zuo [1,2], Yingchao Du [2,3], Lianqi Wei [2], Bo Yu [2,3], Shufeng Ye [2], Xiaomeng Zhang [2,*] and Hongshun Hao [1,*]**

[1] School of Textile and Material Engineering, Dalian Polytechnic University, No. 1 Light Industry Court, Ganjingzi District, Dalian 116034, China; ZuoGaohong@163.com

[2] State Key Laboratory of Multiphase Complex Systems, Institute of Process Engineering, Chinese Academy of Sciences, PO Box 353, Beijing 100190, China; ycdu@ipe.ac.cn (Y.D.); lqwei@ipe.ac.cn (L.W.); yubo@ipe.ac.cn (B.Y.); sfye@ipe.ac.cn (S.Y.)

[3] Center of Materials Science and Optoelectronics Engineering, University of Chinese Academy of Sciences, No. 19(A) Yuquan Road, Beijing 100049, China

[*] Correspondence: xmzhang@ipe.ac.cn (X.Z.); beike1952@163.com (H.H.)

**Abstract:** Emulsified pickling waste liquid, derived from cleaning oily hardware, cause serious environmental and ecological issues. In this work, a series of grafted (3-aminopropyl)triethoxysilane (APTES) TiO$_2$ pillared montmorillonite (Mt), Ti-Mt-APTES, are prepared and characterized for their assessment in demulsification of acidic oil-in-water emulsion. After titanium hydrate is introduced through ion exchange, montmorillonite is modified by hydrophobic groups coming from APTES. The Ti-Mt-APTES in acidic oil-in-water emulsion demulsification performance and mechanism are studied. Results show that the prepared Ti-Mt-APTES has favorable demulsification performance. The Ti-Mt-APTES demulsification efficiency (E$_D$) increased to an upper limit value when the mass ratio of APTES to the prepared TiO$_2$ pillared montmorillonite (Ti-Mt) (R$_{A/M}$) was 0.10 g/g, and the 5 h is the optimal continuous stirring time for breaking the acidic oil-in-water emulsion by Ti-Mt-APTES. The E$_D$ increased to 94.8% when 2.5 g/L of Ti-Mt-APTES is added into the acidic oil-in-water emulsion after 5 h. An examination of the demulsification mechanism revealed that amphiphilicity and electrostatic interaction both played vital roles in oil-in-water separation. It is demonstrated that Ti-Mt-APTES is a promising, economical demulsifier for the efficient treatment of acidic oil-in-water emulsions.

**Keywords:** modified montmorillonite; oil-in-water emulsion; acidic demulsification; silane couple grafting

## 1. Introduction

It has been recognized that surface treatment is a crucial aspect of metalworking processes [1]. Before drawing, spraying, and electroplating of iron and steel, a pickling solution is normally used to remove rust, dust, and surplus parts on the surface of iron and steel [2]. Hydrochloric acid pickling has the advantages of having the highest pickling capacity, a fast speed, and no oxygen embrittlement. Hence, hydrochloric acid is the most widely used pickling medium [3]. Chinese steel enterprises discharge more than 1 million tons of pickling waste liquid every year [4], containing a large amount of hydrogen ions and emulsified oils [5]. The particularity of the water quality of pickling waste liquid determines its harmfulness to the water environment and ecology [6]. Pickling of the metal surface will cause a large amount of emulsified oils to enter the cleaning solution, and excessive emulsified

oils in the cleaning solution are an important factor leading to the scrapping of the pickling solution. Hence, separating emulsified oils from emulsified pickling solution can prolong the service life of the pickling solution, which leads to a reduction in the discharge of pickling waste liquid.

Commonly, traditional methods of separating oil from oil-in-water or water-in-oil emulsions contain flotation [7], coagulation [8], ultracentrifugation, chemical and electrochemical demulsification [9], gravity separation [10], and membrane filtration [11]. Due to its high efficiency, easy operation, and low-energy consumption, chemical demulsification is a practical method for treating oil-in-water emulsions [12]. In addition, chemical demulsifiers can expedite the emulsion-breaking process, and their application conditions are neutral or weakly alkaline. So far, multitudinous chemical demulsifiers are suggested for demulsification, such as cationic polymers, non-ionic block polymers, and dendrimers [13,14]. As shown in previous papers, these demulsification performances are sensitive to pH, and especially poor under acidic conditions [15].

Montmorillonite is a natural material with a layered structure, enormous surface area, high cation exchange capacity, and can absorb organic substances [16]. Nowadays, the preparation and application of montmorillonite and related products principally focus on environmental pollution improvement [17,18], nanocomposites [19], and catalyst carriers [20]. As for application of montmorillonite in environmental pollution, montmorillonite is primarily studied for the adsorption of heavy metal ions and organic pollutants.

However, there have been limited studies on montmorillonite demulsification in emulsified pickling waste liquid [21]. Montmorillonite is relatively stable under acidic conditions [22] and organically modified to be introduced hydrophobic groups, making the clay amphiphilic. Therefore, organically modified montmorillonite can be used in an acidic environment. The demulsifiers that are being studied are almost never used in acidic conditions, especially in strong acid conditions. The idea that montmorillonite demulsifies in an acidic environment is novel. Organically modified montmorillonite achieves separation of the emulsified oil from the emulsion in an acidic environment, which can prolong the service life of pickling liquid and reduce the discharge of pickling waste liquid in the metal pickling process.

## 2. Materials and Methods

### 2.1. Material

Raw montmorillonite (Mt) with a specific surface area and mean grain size of 49.23 $m^2/g$ and 12.55 μm, respectively, was obtained from Adamas Reagents. Sodium montmorillonite (Na-Mt) was prepared by treatment of Mt with a 1.0 M sodium chloride solution. The cation exchange capacity (CEC) of Na-Mt was 100 mmol/100 g, which was calculated using the ammonium chloride–50% ethanol method [23]. Titanium(IV) isopropoxide (98%), (3-aminopropyl)triethoxysilane (APTES, 98%), ethanol, and sodium chloride (NaCl) were supplied by Aladdin Chemistry (Shanghai, China). Polyoxyethylene (80) sorbitan monooleate (Tween 80, $C_{24}H_{44}O_6$) was provided by Xilong Chemical Co., Ltd. (Beijing, China) Hydrochloric acid (HCl) was purchased from Beijing Xingqing Red Fine Chemical Technology Co., Ltd. (Beijing, China). Deionized water was used throughout the experiment. All chemicals were directly used without further treatment.

### 2.2. Preparation of Ti-Mt-APTES

The sol–gel method was adopted to prepare $TiO_2$ pillared montmorillonite (Ti-Mt) [24]. A titania sol was obtained by hydrolysis of titanium tetraisopropoxide with a molar ratio of $Ti(OC_3H_7)_4/HCl$ of 1/4. By mixing the Na-Mt suspension (1 wt %) with titania sol with a molar ratio of Ti/CEC at 20, the Ti-Mt wet powders were then ready for drying.

Various mass of APTES was added to the mixture of Ti-Mt and ethanol with the ratio of 1/1 (w/v) under vigorous mechanical stirring at 80 °C for 5 h. The resulting product was centrifuged and dried under vacuum at 60 °C for 12 h. The mass ratio of APTES to the prepared Ti-Mt

($R_{A/M}$) was determined to be between 0 and 0.12 g/g. The Ti-Mt-APTES samples obtained at an $R_{A/M}$ of 0, 0.02, 0.04, 0.06, 0.08, 0.10, and 0.12 g/g were denoted as Ti-Mt, Ti-Mt-APTES-0.02, Ti-Mt-APTES-0.04, Ti-Mt-APTES-0.06, Ti-Mt-APTES-0.08, Ti-Mt-APTES-0.10, and Ti-Mt-APTES-0.12, respectively. The preparation of Ti-Mt-APTES is depicted in Figure 1.

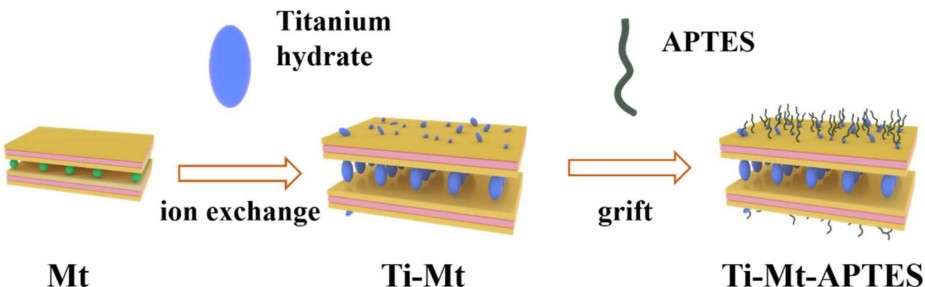

**Figure 1.** Preparation of Ti-Mt-APTES. Mt. Mt, montmorillonite; APTES, (3-aminopropyl)triethoxysilane.

### 2.3. Demulsification Tests

Tween 80 (0.25 g) was added, as an emulsifier, to a mixture of 100 mL 1 M HCl and 2 g diesel (a mixture of alkanes, alkenes, naphthenes, aromatic hydrocarbons, polycyclic aromatic hydrocarbons, and additives). After stirring at 1200 r/min for 4 h, the acidic oil-in-water emulsion was obtained. A given amount (0.02~0.06 g) of clays and 20 mL of freshly prepared acidic oil-in-water emulsion were thoroughly mixed in a 100 mL beaker. The mixture was agitated in a DF-101S thermostatic water bath shaker for various times at 25 °C. Then, the mixture allowed to stand 1 h, and residual oil content of the liquid phase was measured by monitoring the absorbance at 570 nm using a UV–visible spectrometer (YUKE, UV752N). The demulsification efficiency ($E_D$) was calculated from the measured residual oil content by

$$E_D = \frac{A_0 - A_r}{A_0} \times 100\% \tag{1}$$

where $A_0$ and $A_r$ were the original and residual oil contents of the liquid phase, respectively [25].

### 2.4. Characterizations

The morphology and elemental distribution of the as-prepared samples were examined by scanning electron microscopy (SEM) and energy dispersive spectrometer (EDS) simultaneously operated at 15 kV (SEM, JSM-7001F, JEOL). The specific surface areas of the materials were calculated by specific surface area and microporous analyzer (BET, SSA-7300, BIAODE, Beijing, China). The X-ray diffractometer (XRD, Panalytical X' pert PRO MPD, PANalytical B.V.) equipped with Cu Kα radiation was employed to detect the chemical compositions of samples. The Fourier transform infrared (FTIR, T27, Bruker) spectroscopy measurements were carried out in KBr pellet at room temperature. The chemical compositions of the material surface were determined by X-ray photoelectron spectra analysis (XPS, ESCALAB 250Xi, Thermo Fisher Scientific, Waltham, MA, USA) using monochromatic Al Kα radiation. The surface zeta potentials of samples were measured using Zetasizer Nano ZS instrument (Delsa Nano C, Beckman Coulter Inc, Brea, CA, USA). The prepared emulsion and the Ti-Mt-APTES demulsification processes were observed and photographed using a microscope (BX51M, Olympus, Tokyo, Japan).

## 3. Results and Discussion

### 3.1. Characterization of Na-Mt, Ti-Mt, and Ti-Mt-APTES

Figure 2 shows the XRD patterns of the Na-Mt and Ti-Mt-APTES. The characteristic peaks for montmorillonite can be clearly seen in all samples, while peaks of quartz and cristobalite were present owing to the introduction of impurities. It can be found that the (001) plane assigned to the parent

Na-Mt (2θ = 6.91°) is shifted toward lower values of 2θ (2θ = 5.92°), due to the intercalation of titanium hydrate and APTES, which lead to the increase of the basal spacing. The crystal surface spacing $d_{(001)}$ of montmorillonite is estimated according to Bragg's law:

$$n\lambda = 2d \sin\theta \tag{2}$$

where [26] the basal spacing ($d_{(001)}$) of Na-Mt is 1.28 nm, and the spacing ($d_{(001)}$) for Ti-Mt-APTES is 1.49 nm. The similarities in the spectra of the two samples indicate no structural changes have occurred in the clay matrix.

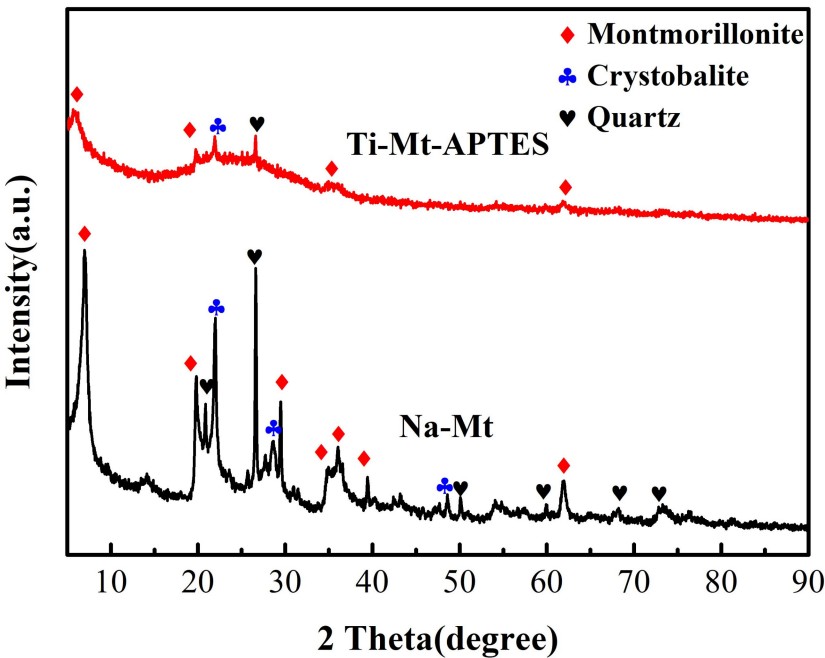

**Figure 2.** XRD patterns of Na-Mt and Ti-Mt-APTES.

The FTIR spectra of Na-Mt and Ti-Mt-APTES are illustrated in Figure 3. For Na-Mt, it is possible to assign the peaks located at 3618 and 1036 cm$^{-1}$ to the Al–O–H and Si–O–Si stretching vibrations [27]. Extra peaks relevant to H–O–H stretching and bending vibrations can be found at 3448 and 1640 cm$^{-1}$, respectively. In the case of Ti-Mt-APTES, additional peaks at 2925 and 2852 cm$^{-1}$ are attributed to antisymmetric [28] and symmetric stretching vibrations of –CH$_2$ [29], signifying that APTES successfully loaded onto montmorillonite.

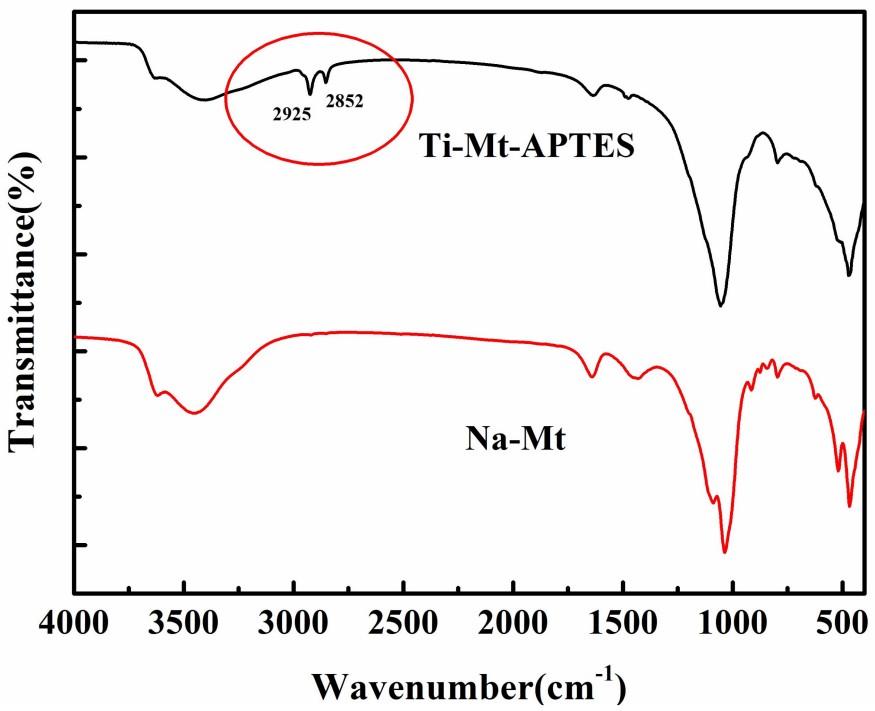

**Figure 3.** FTIR spectra of Na-Mt and Ti-Mt-APTES.

The nitrogen adsorption/desorption isotherms of Na-Mt and Ti-Mt-APTES are displayed in Figure 4. As can be seen from curves, Na-Mt demonstrates type III isotherms on the base of the BDDT classification, showing macroporous adsorbents [30], while the Ti-Mt-APTES curve is characteristic for type IV solids, indicating mesoporous structures [31]. The specific surface areas of Na-Mt and Ti-Mt-APTES are calculated to be 45.14 and 190.20 m$^2$/g, respectively. The Ti-Mt-APTES specific surface areas increased, indicating the materials have strong adsorption capacity, which might be ascribed to intercalation of titanium hydrate and grafted APTES.

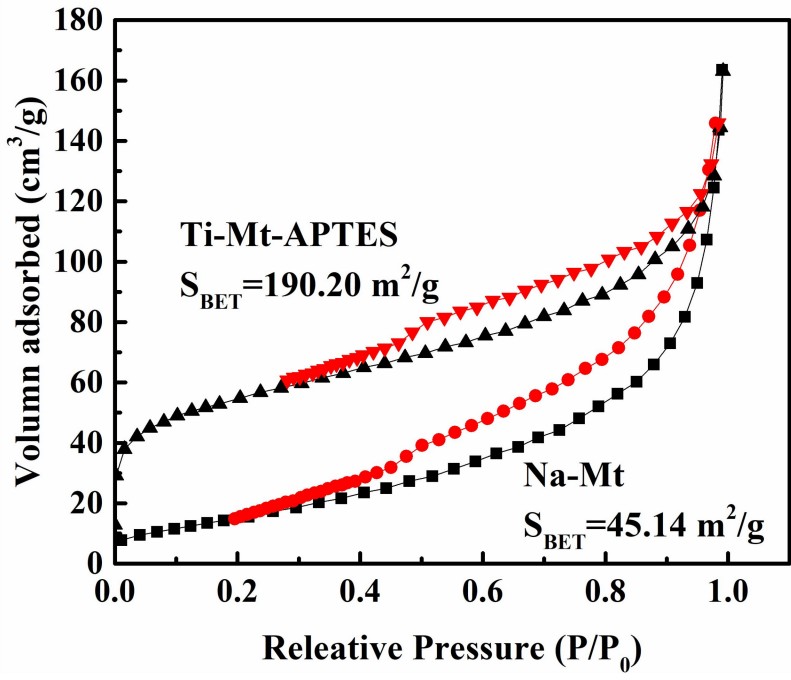

**Figure 4.** N$_2$ adsorption/desorption isotherms of Na-Mt and Ti-Mt-APTES.

The typical SEM images of Na-Mt and Ti-Mt-APTES are observed in Figure 5a–c. The Na-Mt micrograph exhibits larger particle aggregates with smooth surfaces. Layer structure of Na-Mt can be clearly observed, and the structure is more compact. However, Ti-Mt-APTES shows relatively disordered and loose layer structures, attributed to titanium hydrate entering the interlayer of montmorillonite and acting as a spacer for the clay layer, and does not destroy the layer structure of montmorillonite. Furthermore, a mass of small aggregates, which are presumably broken platelets and agglomerates of titanium hydrate, result from hydrolysis of surface $Ti(OC_3H_7)_4$ [32]. The EDS mapping of Ti-Mt-APTES is given in Figure 5c. This clearly indicates that C, N, and Ti elements in the Ti-Mt-APTES are evenly distributed, suggesting the successful intercalation of titanium hydrate and grafted APTES. The above results verify Figure 1.

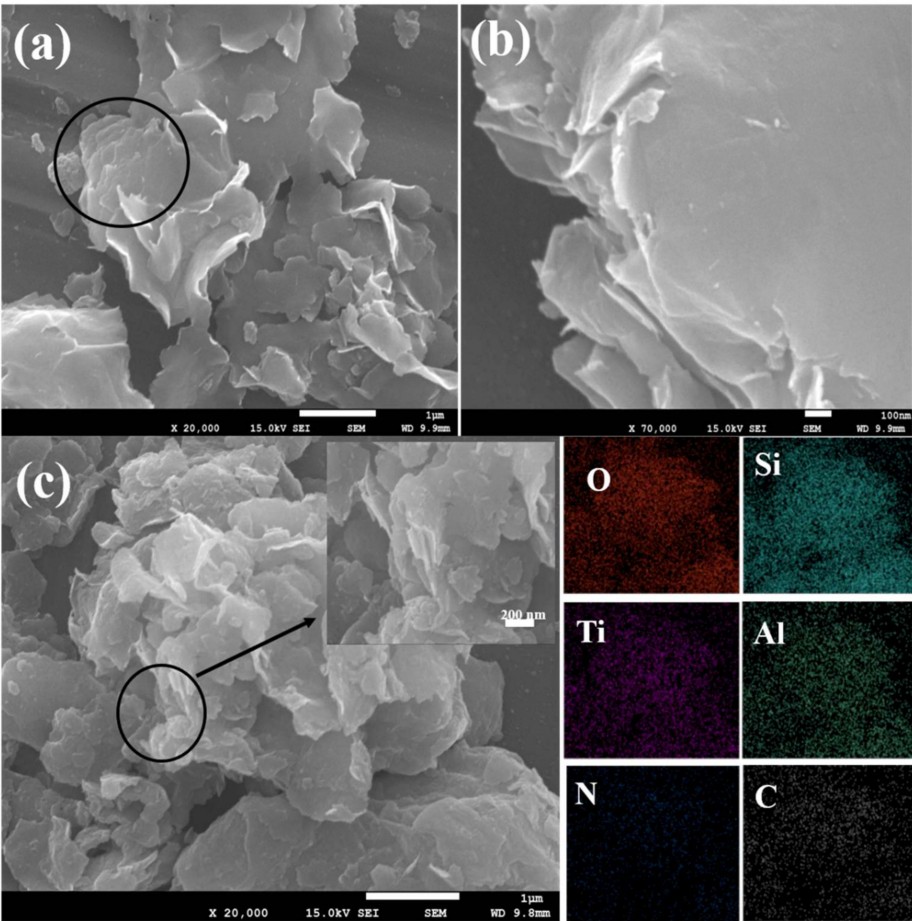

**Figure 5.** SEM image of Na-Mt (**a**), circle enlarged image (**b**), SEM image of Ti-Mt-APTES (**c**) and EDS results of Ti-Mt-APTES.

### 3.2. Demulsification Performance of Ti-Mt-APTES.

Demulsification performance is evaluated by using the absorbance of emulsion before and after treating acidic oil-in-water emulsions, and observed intuitively by color change of emulsions. Photographs of the acidic oil-in-water emulsion and after demulsification by Ti-Mt-APTES-0.10 are shown in Figure 6. This clearly shows that the acidic oil-in-water emulsion is a uniform milky liquid, while the mixture shows three parts after demulsification by Ti-Mt-APTES-0.1: the uppermost part is oil phase, the middle part is water phase, and the bottom part is modified montmorillonite. The results demonstrate that Ti-Mt-APTES-0.10 composite is an effective demulsifier to separate oil from acidic oil-in-water emulsions.

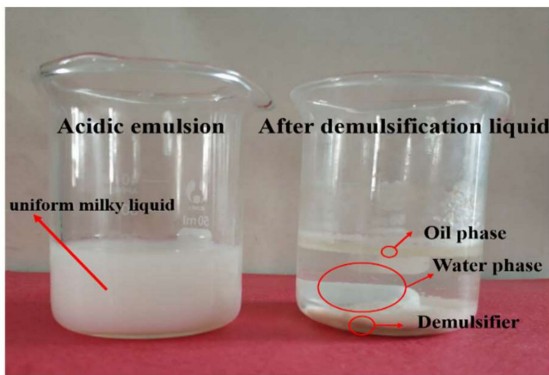

**Figure 6.** Photographs of the acidic oil-in-water emulsion and after demulsification by Ti-Mt-APTES-0.10.

Figure 7a clearly indicates that pure Mt and Ti-Mt $E_D$ are 2.67% and 3.17%, respectively, at the dosage of 2.5 g/L, and show almost no demulsification performance while, upon anchoring APTES, the demulsification efficiency of Ti-Mt and Na-Mt can be seen to significantly improve in acidic conditions. As shown in Figure 7b, the Ti-Mt-APTES demulsification performance increases with the increase of APTES-anchoring densities, but when it exceeded 0.10g/1g, the demulsification ability decreased, which is likely to decrease the active hydroxyl on the materials. Furthermore, the dosage of Ti-Mt-APTES also has an important effect on the $E_D$ of acidic oil-in-water emulsions and, as the dosage of Ti-Mt-APTES increases, the performance of demulsification increases, which is attributed to an increase in the addition of Ti-Mt-APTES, which will provide more active sites for adsorbing oil droplets. As for Ti-Mt-APTES-0.10, the $E_D$ was 94.8% at a dosage of 2.5 g/L. In addition, it can be concluded that 0.10 g/1 g of $R_{A/M}$ is the optimal condition for separating oil from an acidic emulsion by Ti-Mt-APTES, and the optimal dosage was 2.5 g/L.

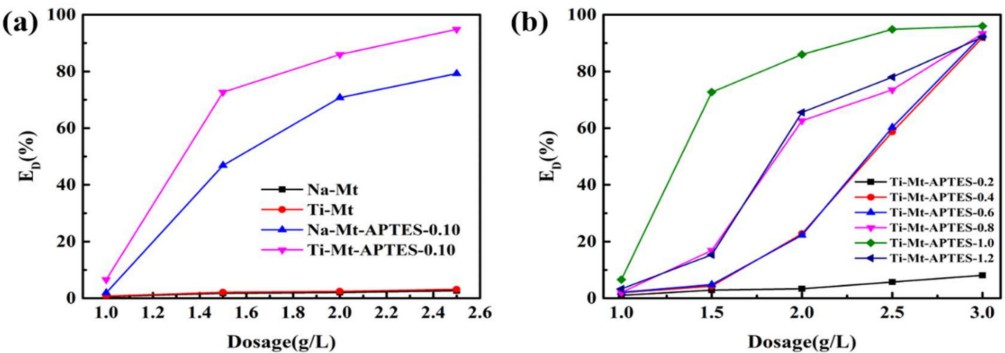

**Figure 7.** (**a**) Demulsification performance of Na-Mt, Ti-Mt, Na-Mt-APTES-0.10, and Ti-Mt-APTES-0.10. (**b**) Effect of the Ti-Mt-APTES dosage on demulsification efficiency at various $R_{A/M}$.

As shown in Figure 8, the stirring time plays a significant role in achieving $E_D$ during the demulsification process. Initially, Ti-Mt-APTES-0.10 shows low $E_D$ in a relatively short period of time, and approximately 6.8% of the $E_D$ at a dosage of 2.5 g/L after 2 h. Subsequently, the $E_D$ increased and reached 94.8% at a dosage of 2.5 g/L after 5 h. The $E_D$ is about 94.9% for Ti-Mt-APTES-0.10 at the same dosage after 6 h. Additionally, it can be concluded that 5 h for the continuous separation of oil from an acidic emulsion is optimal for breaking the emulsion using Ti-Mt-APTES-0.10.

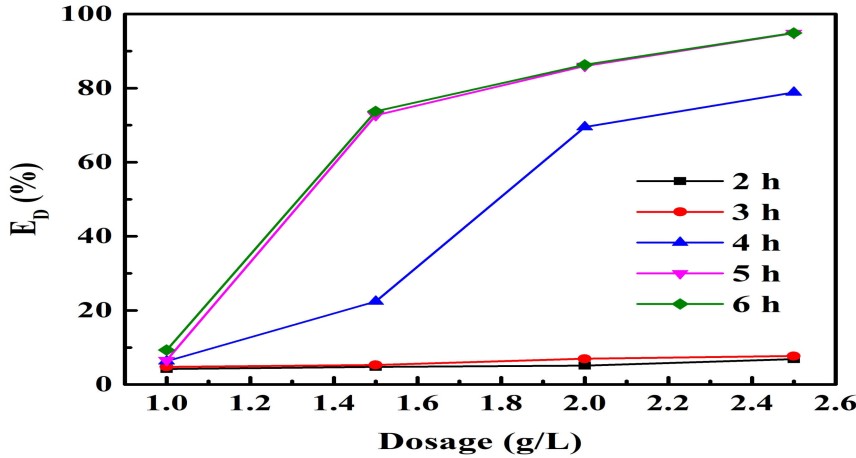

**Figure 8.** Demulsification performance of Ti-Mt-APTES-0.1 under various stirring times.

## 4. Demulsification Mechanism

To further comprehend the demulsification process of the acidic oil-in-water emulsion actuated by Ti-Mt-APTES, the morphologies of the emulsions were observed with a metallographic microscope. Figure 9a shows the optical morphology of the acidic oil-in-water emulsion, indicating the oil droplets homogeneously dispersed in the water phase. As shown in Figure 9b, after demulsification, a small amount of diminutive oil droplets appeared in the newly formed water phase. As for the separated oil phase, there are bits of Ti-Mt-APTES floccules suspended in the oil phase, and a small quantity of large water droplets are also present (Figure 9c). Moreover, floccules of the Ti-Mt-APTES are also found at the oil/water interface, as collected large oil droplets (Figure 9d). These results demonstrate that Ti-Mt-APTES is an efficient demulsifier to separate oil from the acidic oil-in-water emulsion.

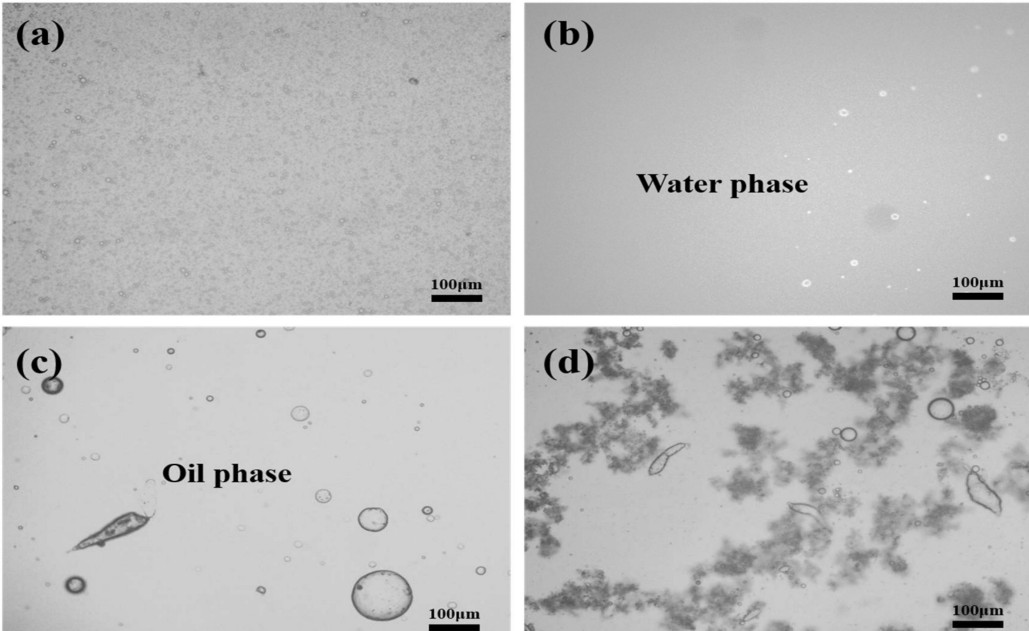

**Figure 9.** Micrographs of the oil–water mixture before and after demulsification: (**a**) the oil-in-water emulsion; (**b**) the newly separated water phase; (**c**) the newly separated oil phase; (**d**) Ti-Mt-APTES in oil-in-water emulsion.

FTIR spectra of Ti-Mt-APTES before and after demulsification are shown in Figure 10. In the case of demulsification, peaks at 2925 and 2852 cm$^{-1}$ strengthened, which correspond to antisymmetric

and symmetric stretching vibrations of –CH$_2$, most likely due to Ti-Mt-APTES catching oil droplets in the acidic oil-in-water emulsion. Furthermore, peaks related to H–O–H stretching and bending vibrations can be found at 3448 and 1640 cm$^{-1}$, and weaken after demulsification, indicating that the hydroxyl groups on the montmorillonite may be involved in demulsification, and the oil droplets may be adsorbed by intercalation of Ti-Mt-APTES. Therefore, when R$_{A/M}$ exceeds 0.10/1, the hydroxyl groups on the surface of montmorillonite are drastically reduced, which makes the demulsification performance lower.

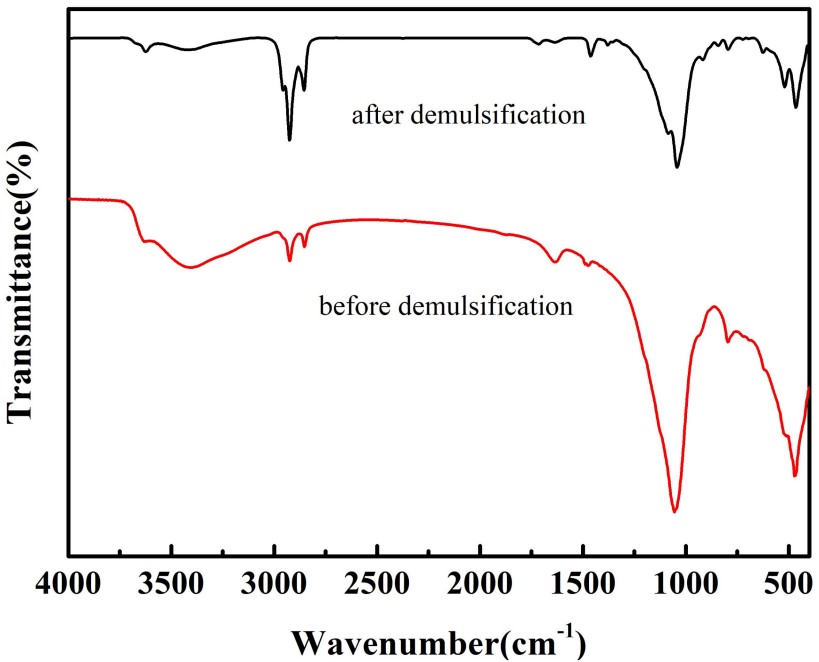

**Figure 10.** FTIR spectra of Ti-Mt-APTES before and after demulsification.

The high resolution XPS spectra of Ti-Mt-APTES before and after demulsification are given in Figure 11. The Ti 2p peaks of Ti-Mt-APTES are located at 458.5, 460.0, and 464.6 eV, respectively, and those after demulsification are located at 465.1, 460.0, and 459.0 eV, respectively, which can be attributed to the Ti$^{4+}$ oxidation sates or TiO$_2$ groups in montmorillonite [33]. The C 1s peaks are observed at 286.6 and 284.7 eV, and after demulsification, C 1s peaks were found at 286.0 and 284.64 eV, which were assigned to elemental carbon (C–C) and C–O [34]. However, those after demulsification are positioned at 286.0 and 284.6 eV. The N 1s peak is found at 401.6 eV and there was no change after demulsification, which indicates that APTES was successfully loaded onto clay [35]. The Ti-Mt-APTES spectra contain one O 1s peak located at 532.0 eV, that after demulsification was centered at 532.5 eV, and is allocated to single-bonded oxygen from the montmorillonite lattice [36]. From above, the peaks in the O 1s and C 1s regions show shifts to lower binding energy values, and the Ti 2p peak regions change. This phenomenon directly confirmed the interaction between oil droplets and Ti-Mt-APTES.

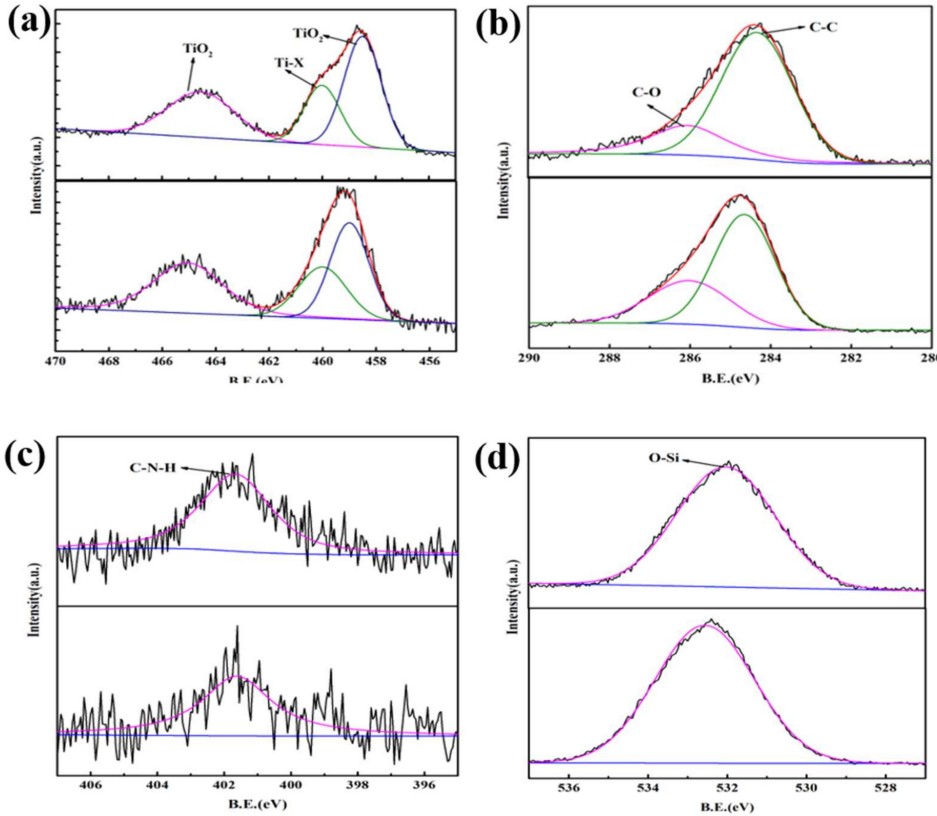

**Figure 11.** High resolution XPS spectra of Ti 2p (**a**), C 1s (**b**), N 1s (**c**), O 1s (**d**) in Ti-Mt-APTES before and after demulsification.

The zeta potentials of Na-Mt and Ti-Mt-APTES at various pH are presented in Figure 12. The zeta potential of Na-Mt remained negative under the measured pH range (pH = 2~8) and decreased with the increase of pH. However, Ti-Mt-APTES was positively charged, with a zeta potential value of 3.62 mV at pH 2.0, ascribed to the protonation of aminopropyl groups and titanium hydrate. By contrast, emulsified oil droplets are negatively charged in acidic emulsions [37]. Therefore, emulsified oil droplets can be efficiently seized on Ti-Mt-APTES by electrostatic interactions in acidic conditions.

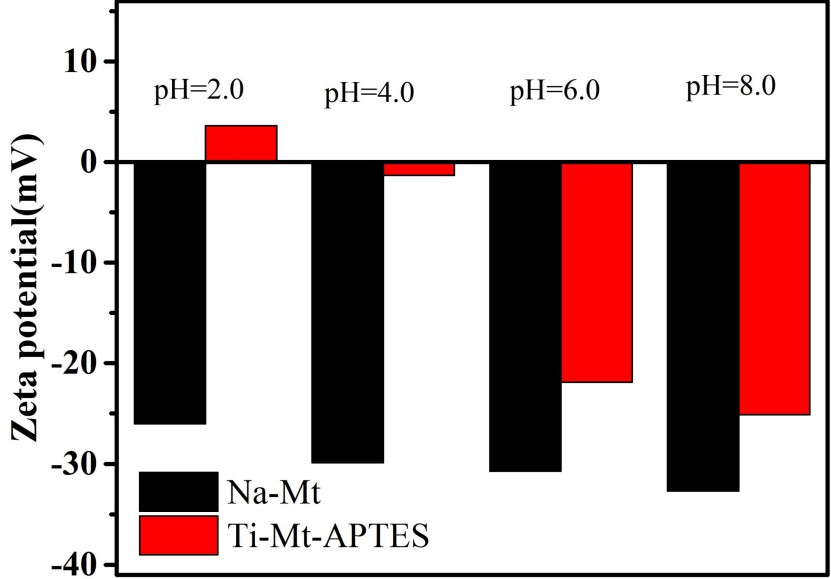

**Figure 12.** Zeta potentials of Na-Mt and Ti-Mt-APTES at various pH levels.

From the above analysis, the mechanism of demulsification is proposed and schematically illustrated in Figure 13. Negatively charged Na-Mt was modified to make the surface positively charged in an acidic environment and has amphiphilic properties. The Ti-Mt-APTES with hydrophilic titanium hydrate and hydrophobic grafted APTES can be well dispersed in the water phase. In acidic oil-in-water emulsions, Ti-Mt-APTES behaves like an amphiphilic flocculant. The seizing of oil droplets by Ti-Mt-APTES is divided into two aspects. On the one hand, in acidic oil-in-water emulsion systems, Ti-Mt-APTES can capture the major available oil droplets with a negative surface by electrostatic interactions, consistent with Figure 12. Moreover, the hydrophobic propyl on montmorillonite tends to adsorb hydrophobic oil droplets via hydrophobic–hydrophobic attraction, as confirmed in Figure 10. On the other hand, intercalation of titanium hydrate can adsorb oil droplets via van der Waals forces, promote oil adsorption from the acidic oil-in-water interface, and further improve emulsion separation. The oil droplets are attached onto Ti-Mt-APTES and prone to coalescing when the clays are close to one another. With the increase of coalescence and aggregation of oil droplets, they fall out and float to the surface, bringing about demulsification.

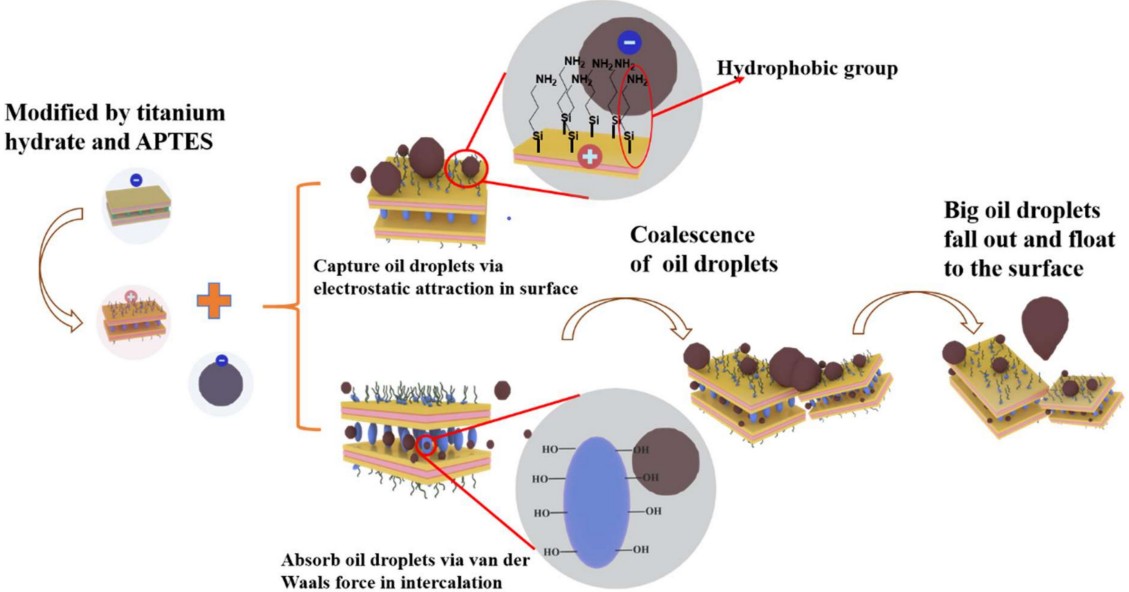

**Figure 13.** Schematic illustration of the demulsification mechanism.

## 5. Conclusions

In this study, a series of Ti-Mt-APTES were prepared for their evaluation in demulsification of acidic oil-in-water emulsions. The effects of Ti-Mt-APTES dosages, $R_{A/M}$, and stirring time of as-prepared Ti-Mt-APTES on oil-in-water separation performance were evaluated. Moreover, the demulsification mechanism was explored.

(1) The $E_D$ of Ti-Mt-APTES-0.10 at acidic conditions is above 94.8% at a dosage of 2.5 g/L after 5 h.

(2) Under acidic conditions, oil droplets could be effectively attached onto Ti-Mt-APTES via electrostatic attraction. The hydrophobic groups located in montmorillonite were implicated in enhancing the demulsification performance.

(3) Intercalation of titanium hydrate gives the material a larger specific surface area, which can adsorb oil droplets via van der Waals force.

**Author Contributions:** Conceptualization, X.Z. and H.H.; Formal analysis, G.Z., Y.B. and Y.D.; Funding Acquisition, S.Y.; Investigation, G.Z., Y.D., L.W., H.H. and X.Z.; Software, G.Z., Y.D. and X.Z.; Supervision, S.Y., H.H. and X.Z.; Validation, X.Z. and H.H.; Data curation, Y.D. and X.Z.; Writing—original draft preparation, G.Z.; Writing—review and editing, G.Z., L.W. and X.Z.

**Funding:** This work was funded by Chinese Academy of Science Key Project [grant number KFJ-STS-QYZD-044] and the "Strategic Priority Research Program" of Chinese Academy of Science [grant number XDA09040102].

**Conflicts of Interest:** The authors declare no conflict of interest.

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
