# Peer review of "Efficient Demulsification of Acidic Oil-In-Water Emulsions with Silane-Coupled Modified TiO2 Pillared Montmorillonite"

_applsci, doi:10.3390/app9051008_

Round 1

Reviewer 1 Report

In my opinion generally the subject of the manuscript is interesting. However, novelty and aim of the work should be clearly stated in Introduction. Additionally, more results on Na-Mt-APTES should be also presented and compared with Ti-Mt-APTES. Also, reuse of the materials should be proven.

The main doubt is about composition of the separated emulsion it seems to be completely different from spent pickling solutions (broadly described in Introduction).

English quality should be improved because some parts are difficult to understand.

More detailed remarks are included in the attached file.

Author Response

Dear reviewer,

Thank you very much for your comments on our manuscript entitled “Efficient demulsification of acidic oil-in-water emulsions with silane-coupled modified TiO2 pillared montmorillonite”. Those comments are all valuable and very helpful for revising and improving our paper, as well as the important guiding significance to our researches. We have studied comments carefully and have made correction which we hope meet with approval. And then, we carry out a comprehensive revision of the manuscript. Revised portion are marked in red in the paper. The main corrections in the paper and the responds to the reviewers’ comments are as follows:

Responses to the review 1 comments:

Comment 1: You write here that there were some studies on use of montmorillonite for demulsification, so you should refer to them. There is no example here.

Response: Thank you very much for the suggestion. The references of montmorillonite for demulsification are cited.

Comment 2: State clearly aim of your work.

Response: Thank you very much for the suggestion. The aim of our work is clearly redefined in the manuscript.

Comment 3: Give details on chemical composition of Tween 80.

Response: Thank you very much for the suggestion. The details on chemical composition of Tween 80 are added.

Comment 4: Why do you use TiO2 pillared montmorillonite, not TiO2 or montmorillonite themselves?

Response: Thank you very much for the suggestion. Pillared montmorillonite has a larger specific surface area, which more likely to adsorb oil droplets. What’s more, TiO2 pillared montmorillonite is a common type of pillared montmorillonite.  

Comment 5: Why do you prepare diesel emulsion? The whole Introduction you write about spent pickling solutions and emulsions formed in them. And here you use completely different solution. You should work with model and real pickling solutions.

Response: Thank you very much for the suggestion. The composition of pickling waste liquid fluctuates greatly, making it difficult to obtain representative samples. What’s more, our prepared acidic diesel emulsion can well cover the common characteristics of on-site pickling waste liquid.

Comment 6: Give more details on methodology of the measurements.

Response: Thank you very much for the suggestion. The details on methodology of measurements are described.

Comment 7: Place also colour lines in the caption to make clear which line stands for the appropriate material.

Response: Thank you very much for the suggestion. We add some text to distinguish the two lines in the Figure 3.

Comment 8: What is stability of the material? Is APTES lost during demulsification?

Response: Thank you very much for the suggestion. From the high resolution XPS spectra of Ti-Mt-APTES before and after demulsification, the N 1s peak has no change after demulsification, which indicate the Ti-Mt-APTES is stable during demulsification. APTES is stably loaded on montmorillonite during demulsification and is not lost.

Comment 9: Mt modified with APTES should also be tested.

Response: Thank you very much for the suggestion. The demulsification performance of Na-Mt-APTES is tested and displayed in Figure 7(a).

Comment 10: Compare this time with other methods of demulsification.

Response: Thank you very much for the suggestion. Most demulsifies application environment are neutral or weakly alkaline, and little demulsifiers are used in strong acid environment. So, little relevant data compare to it.  

Comment 11: You should also study Na-Mt-APTES. Maybe it is not necessary to apply additional operation of Ti introducing into Mt. What is role of Ti in this material? Maybe it would be enough to use only APTES without solid material?

Response: Thank you very much for the suggestion. The demulsification performance of Na-Mt-APTES have been studied. Compare with Ti-Mt-APTES, it demulsification efficiency is relatively low at the same conditions. So, it is necessary to modify montmorillonite by TiO2 pillar.

Comment 12: Can this material be reused? Please present the results for reusing and yield of demulsification with reused material.

Response: Thank you very much for the suggestion. The recyclability of the material has been studied, and displayed in the following figure. The material can be reused, but the demulsification performance decreases a lot when the material circulates for a certain number of times. The material recovery cost is relatively high, but it is cheap and easy to get. So, this material recycling is of little significance.

In all, we found the Reviewers’ comments are quite helpful, and we have revised the paper carefully. Thank you and the review again for your help!

Should you have any questions about this paper, please let us know without hesitation.

  Thank you and best regards.

  Yours sincerely,

Reviewer 2 Report

paragraph 2.1. Material

General note: Could be usefull to add a preliminary characterization of the raw material as received. (i.e. Montmorillonite  specific area, pH, grain diameter...)

line 73 Please Correct the typo  "throufhout" instead of "throughout" 

paragraph 2.2. line 78 please clarify what "Ti/CEC" stand for

paragraph 3.2. Demulsification performance of Ti-Mt-APTES

line 158 it is not clear the observation conditions (when the two picture were taken?) Could be usefull to  add the picture of the other solution to compare the results?

line 168 Can you please clarify the mechanism behind the dosage of the Ti-Mt-APTES on the ED of the acidic oil-in-water emulsions

Author Response

Dear reviewer,

Thank you very much for your comments on our manuscript entitled “Efficient demulsification of acidic oil-in-water emulsions with silane-coupled modified TiO2 pillared montmorillonite”. Those comments are all valuable and very helpful for revising and improving our paper, as well as the important guiding significance to our researches. We have studied comments carefully and have made correction which we hope meet with approval. And then, we carry out a comprehensive revision of the manuscript. Revised portion are marked in red in the paper. The main corrections in the paper and the responds to the reviewers’ comments are as follows:

Responses to the review 2 comments:

Comment 1: Could be usefull to add a preliminary characterization of the raw material as received. (i.e. Montmorillonite specific area, pH, grain diameter...)

Response: Thank you very much for the suggestion. The preliminary characterizations of raw montmorillonite are added.

Comment 2: Please clarify what "Ti/CEC" stand for

Response: Thank you very much for the suggestion. "Ti/CEC at 20" means titania sol required for the reaction was 20 times the CEC equivalent of the montmorillonite.

Comment 3: It is not clear the observation conditions (when the two pictures were taken?) Could be usefull to add the picture of the other solution to compare the results?

Response: Thank you very much for the suggestion. The details of pictures are added in Figure 6, which clearly shows when the two photos were taken.

Comment 4: Can you please clarify the mechanism behind the dosage of the Ti-Mt-APTES on the ED of the acidic oil-in-water emulsions

Response: Thank you very much for the suggestion. The mechanism behind the dosage of the Ti-Mt-APTES on the ED of acidic oil-in-water emulsions is depicted.

Response to other comments: We corrected the spelling and grammar mistakes.

In all, we found the Reviewers’ comments are quite helpful, and we have revised the paper carefully. Thank you and the review again for your help!

Should you have any questions about this paper, please let us know without hesitation.

  Thank you and best regards.

  Yours sincerely,

Round 2

Reviewer 1 Report

The manuscript is improved but still some remarks are treated superficially. I still insist on some additional information or changes. Without them I do not recommend the article for publication.

These are:

Introduction, line 37: Clearly write what "oils" you mean, give some examples. Define this "oily" compositon of Chinese spent pickling solutions.

Introduction, line 59: Ref. 21 is unavailable, it cannot be found and read. Ref. 22 has nothing to do with pickling waste water. You should give examples of investigation on montmorillonite application for emulsified pickling solutions, to prove your statement.

The last sentence in Introduction still does not clearly state the aim of your work. Write clearly what did you want to achieve or prove by the studies you did. And what is novelty of your investigation compared to the previously carried by others.

Material: You should give full chemical name of the surfactant in Tween 80.

2.3. Deemulsification tests: I am not satisified with answer about use of "diesel". Define this "diesel", what is its chemical composition? In my opinion it has nothing to do with composition of real spent pickling solutions. Give data on composition of a real pickling solution to which you refer adding this diesel. What is a source of diesel in acidic pickling solutions? In my opinion application of diesel is not justified.

Please introduce information on recyclability and proposal of treatment of waste Ti_Mt_APTES into your manuscript.

In response you write that "The material recovery cost is relatively high, but it is cheap and easy to get. So, this material recycling is of little significance." But if you do not recycle it, you produce solid waste. So what is your idea to manage waste Ti_Mt_APTES? Remember that you used diesel emulsion so this is environmentally unfriendly material.

Author Response

Dear reviewer,

Thanks very much for your comments on our manuscript entitled “Efficient demulsification of acidic oil-in-water emulsions with silane-coupled modified TiO2 pillared montmorillonite”. Those comments are all valuable and greatly helpful for revising and improving our paper, as well as the important guiding significance to our researches. We have studied comments carefully and have made correction which we hope meet with approval. And then, we have further revised the manuscript. Revised portion are marked in red in the paper. The main corrections in the paper and the responds to the reviewers’ comments are as follows:

Responses to the review comments:

Comment 1: What "oils" do you mean, give some examples? Define this "oily" composition of Chinese spent pickling solutions.

Response: Thank you very much for the suggestion. The “oils” means oils on the surface of the steel parts, such as lubricating oil and hydraulic oil on the hot rolled plate. Emulsified oil is composed of a variety of oils, mainly mineral oil on the surface of steel materials during processing and use.

Comment 2: This article has nothing to do with pickling waste water. You should give examples of investigation on montmorillonite application for emulsified pickling solutions, to prove your statement.

Response: Thank you very much for the suggestion. The article we quoted is about the demulsification of bentonite. Montmorillonite is similar to the basic composition of bentonite, and there is a high degree of correlation between them. In fact, we have not found any articles about demulsification in acidic environment, especially in strong acid environment. The study on the demulsification of bentonite can provide reference for montmorillonite.

Comment 3: Still there is no clearly stated aim of your work. Write clearly what did you want to achieve or prove by the studies you did. And what is novelty of your investigation compared to the previously carried by others.

Response: Thank you very much for the suggestion. The aim of our work is redefined in the manuscript. Organically modified montmorillonite can be used in an acidic environment. The demulsifiers that are being studied are almost never used in acidic environments especially in strong acid environments. The idea that montmorillonite demulsifies in an acidic environment is novel. Organically modified montmorillonite achieves to separate the emulsified oil in an acidic environment from the emulsion, which can prolong the service life of pickling liquid and reduce the discharge of pickling waste liquid in metal pickling process.

Comment 4: You should give full chemical name of the surfactant.

Response: Thank you very much for the suggestion. The full chemical name of Tween 80 was added.

Comment 5: Define this "diesel", what is its chemical composition? In my opinion it has nothing to do with composition of real spent pickling solutions. Give data on composition of a real pickling solution to which you refer adding this diesel. What would be source of diesel in acidic pickling solutions?

Response: Thank you very much for the suggestion. Diesel is mainly composed of a mixture of alkanes, alkenes, naphthenes, aromatic hydrocarbons, polycyclic aromatic hydrocarbons and additives. The chemical composition of diesel is added. The main compositions of pickling waste liquor are 10-50g/L free hydrochloric acid, 60-250g/L iron, zinc, lead, chromium and other trace heavy metal elements (total about 500mg/L), water and emulsified oil. The emulsified oil is composed of a variety of oils, mainly mineral oil. Diesel is a typical mineral oil. The prepared demulsifiers are used for real pickling waste liquid, and the visual photographs of the real pickling waste liquid before and after demulsification displayed in the following figure. The prepared demulsifiers are good way to separate emulsified oil from real pickling waste liquid.

Photographs of the real pickling waste liquid and after demulsification by prepared demulsifiers.

Response to other comments: We corrected the spelling and grammar mistakes.

Special thanks to you for your good comments.

At last, thanks again for your letter, we hope our corresponding revisions and responses are suitable. Please feel free to contact us at any time if you have any questions or need additional information.

  Thank you and best regards.

  Yours sincerely,

Round 3

Reviewer 1 Report

Thank you for your responses. I have few editing remarks in newly added novelty statement in Introduction and about spaces between numbers and the units in Experimental part.
